# Influence of the Cutting Strategy on the Temperature and Surface Flatness of the Workpiece in Face Milling

**DOI:** 10.3390/ma13204542

**Published:** 2020-10-13

**Authors:** Lukasz Nowakowski, Michal Skrzyniarz, Slawomir Blasiak, Marian Bartoszuk

**Affiliations:** 1Department of Manufacturing Engineering and Metrology, Kielce University of Technology, 25-314 Kielce, Poland; lukasn@tu.kielce.pl (L.N.); mskrzyniarz@tu.kielce.pl (M.S.); 2Department of Manufacturing Engineering and Automation, Opole University of Technology, 45-758 Opole, Poland; m.bartoszuk@po.edu.pl

**Keywords:** face milling, cutting strategy, heat transfer, temperature distribution, FEM

## Abstract

This article analyzes the temperature data obtained for an aluminum alloy face milled using four different cutting strategies. The workpiece temperature was measured at six points with K-type thermocouples. The heat transfer taking place in the cutting zone was also simulated numerically using the finite element method (FEM) and the finite difference method (FDM). The calculation results concerning the distribution of temperature on the workpiece surface were compared with the experimental data. The numerically simulated distribution of temperature on the workpiece surface after face milling was considered in relation to the surface flatness. The findings suggest that the flatness deviations at the workpiece ends were dependent on the depth of cut. Another reason was the cutting strategy selected for the specific thermophysical properties of the workpiece material. Measurement of the workpiece temperature is extremely important because of the thermoelastic behavior and thermal expansion of the material. The isotropic properties of the aluminum alloy make it expand in all directions during milling.

## 1. Introduction

Machining processes, also known as material removal processes, are commonly used in manufacturing. Their main application is to produce a variety of mechanical components. Machining involves the use of a cutting tool to remove the necessary amount of material from a workpiece to achieve the required geometry and dimensions. It is a complex process with a number of physical phenomena occurring at the cutting tool−workpiece interface. When a mathematical model of the process is created, it is essential to take into account the fact that the tool−workpiece system is subjected to compressive, tensile and shear stresses resulting in elastic and plastic deformations as well as friction, which is the main source of heat generated in this zone. All machining processes involve the generation, transfer, storage and release of large amounts of thermal energy. The amount of heat generated in face milling can be high because of the shape of inserts used. It should be noted that almost all cutting power delivered to the system is transferred into heat [1,2,3]. Since face milling is often used as the finishing operation, it is responsible for the surface texture and properties. The research described in this article aimed to determine how the process temperature affected the geometrical errors in various milling strategies. The material tested was one of the most popular aluminum alloys used in industry, especially the automotive and aviation sectors.

Over the years, much research has been devoted to the effects of cutting conditions on the temperature in the tool−workpiece system, especially in turning and milling. The experimental studies on turning include that by Kusiak et al. [4], who investigated how coatings on the cutting tool affect the temperature of steel workpieces. The measurement of temperature in turning and drilling was also discussed in [5], where the researchers analyzed the cutting temperature using thermocouples placed closed to the cutting zone.

Temperature-related problems encountered in multiedge milling are dealt with, for instance, in [6,7,8]. Armendia et al. examined how high cutting speeds in milling contributed to the tool wear. They pointed out that milling differs from other machining processes because of the use of multiedge tools. They investigated the cyclic changes in temperature and stresses accompanying them. To determine the temperature in interrupted-cut milling, they applied microthermal imaging. Abdelkrim et al. analyzed the effect of milling conditions on cutting temperature and residual stress. Sun et al. examined how the tool temperature and workpiece temperature were dependent on the cutting parameters in milling.

Various studies have assessed the influence of the cutting temperature on the service life of cutting tools. Too much heat generated in the cutting zone may lead to excessive tool wear, which means a drop in the cutting efficiency, or even tool damage [9]. High temperature has a negative impact not only on the tool but also on the workpiece. A large amount of heat transferred during cutting to the workpiece may affect its dimensions, geometry and surface quality, as well as material structure, with all these contributing to the working properties of the final product [10]. It can thus be concluded that the heat-related phenomena taking place in the cutting zone have an impact on the whole cutting process, including the life and performance of the tool, the dimensional and geometrical accuracy of the product, its material structure and also the surface finish [1,2,5,11,12,13]. The heat transferred through the tool, the workpiece and the chips is released to the surrounding air or cutting fluid [10,14]. Changes in the cutting speed lead to changes in the amount of heat removed through the chip from the cutting zone. The higher the cutting speed, the higher the percentage of heat removed through the chip. Thus, the chip shape, the chip cross-section and the mechanisms of chip formation and removal are important factors affecting the temperature in the cutting zone.

The literature includes studies aiming to simulate [15,16] and optimize [17] the cutting process so that the excess heat generated at the tool−workpiece interface was reduced. Some research, e.g., [16], was based on finite difference analysis (FDA) to predict changes in the temperature and thermal resistance at the tool−chip interface.

Thermoelastic deformations, causing dimensional and geometrical errors in longitudinal turning, were discussed by Puls in [18]. In [19], Deppermann and Kneer showed how cutting parameters could minimize the heat load on the workpiece. In the study described in [20], FEM simulations were used to calculate the distortions of the workpiece surface and a mathematical model was developed to determine the heat flux at the interface.

From the literature review, it is evident that there has not been much research into the influence of thermal expansion on the dimensional and geometrical accuracy in machining. This article aims to show that dimensional and geometrical errors in milling may be linked to thermoelasticity.

The amount of heat generated in the cutting zone is largely dependent on the cutting parameters, such as the cutting speed, feed rate, depth or width of cut, workpiece geometry, workpiece material, tool material, wrap angle and the cutting strategy. Determining the cutting strategy usually entails optimizing the path the tool needs to pass to remove the required amount of material or controlling the time of contact between the tool and the workpiece. Taking the above into consideration, the authors ran a number of tests to establish to what extent the milling strategy might affect the distribution of heat in the cutting area and consequently the dimensional and geometrical accuracy of the products milled. Manufacturers of cutting tools generally do not provide suggestions concerning the depth of cut and other parameters. It is the role of engineers to select the right depth of cut and other parameters to obtain parts with the required dimensions and geometry. The choice of the depth of cut is particularly important both in roughing and finishing operations as it affects the whole machining process. Sometimes no coolant or compressed air is used, which is the case analyzed here.

The main purpose of the study was to determine how the source of heat in milling affects the distribution of temperature in the workpiece and its thermoelastic deformation. Another objective was to establish whether and to what extent the surface quality was dependent on the machining strategy.

## 2. Experimental Procedure

The aim of the tests was to determine the effects of the cutting strategy on the workpiece temperature in face milling performed on a DMG DMU-50 vertical machining centre (DMG Mori Seiki Co., Ltd., Nagoya, Japan). The tool used for the purpose was a Sandvik Coromant R245-080Q27-12M face milling cutter (Sandvik Coromant, Sandviken, Sweden), 80 mm in diameter, with 245-12T3M-PL4230 45° carbide inserts. The experiments involved removing a certain amount of material, i.e., a layer with a predetermined thickness and width, by using four different milling strategies. All the tests were carried out under the same conditions, with f_z_ = 0.05 mm/tooth, and accordingly, v_f_ = 444 mm/min. The rotational speed of the spindle was 1481 rev/min. The workpieces (75 mm × 30 mm × 33 mm) were made of AlCu4MgSi (EN AW-2017) aluminum alloy. Some of the mechanical and thermophysical properties of the material are provided in Table 1.

Before each test, the workpiece face was ground longitudinally so that the surface cut was parallel to the bottom. This ensured a constant depth of cut throughout the test. The workpiece temperature was measured during milling by means of six K-type thermocouples, each 1 mm in diameter, fitted at a depth of 15 mm beneath the surface of the workpiece. Figure 1 shows the tool−workpiece system with the thermocouples. Thermoconductive silicone paste was applied to enhance the heat transfer between the workpiece and the thermocouples.

The milling was performed using four different strategies.

In strategy 1 (Figure 2), a 3 mm thick layer was removed from the workpiece along its length and width using one pass. Strategy 2 (Figure 2) required two passes of the tool to remove the same volume of the material, and each time the depth of cut, a_p_, was 1.5 mm. Between the passes, the tool had no contact with the workpiece. The idle time comprised the time of tool retracting, the time of tool approaching and the time of tool setting. In strategy 3 (Figure 2), the desired amount material was removed in three identical passes, each to a depth of 1 mm. In strategy 4, six passes were run, each to a depth of cut of 0.5 mm (Figure 2).

The tests aimed to register the cutting temperature by thermocouples and analyze its influence on the values and distribution of surface flatness. The experimental results were compared with the numerically calculated data. The simulations were carried out using the finite difference method [16].

The numerical simulation programs were developed by incorporating the equations governing the heat transfer phenomena. In this case, the source of heat is the mechanical work the tool does during cutting. In the calculations, it was assumed that the flux of heat transferred from the tool to the workpiece material was at 1200 kW/m^2^.

## 3. Numerical Simulation

The distribution of heat in the workpiece was simulated numerically using the modified finite difference method. The calculations were designed to determine the time-varying and temperature-dependent thermophysical properties of the workpiece material. The calculation procedure was based on the algorithm shown in Figure 3. The calculations required creating a mesh of square elements, each with an edge length of 0.0025 mm. The tool volume was not taken into account. It was assumed that the source of heat with known intensity moved across the workpiece surface introducing a certain amount of heat into it. The geometrical dimensions of the calculation model with the heat source are shown in Figure 4. The length of the flat uniform source of heat was determined analytically on the basis of the stereometry of the milling cutter edges and two cutting parameters f_z_ and a_p_. The method for determining the length of the flat uniform source of heat is illustrated in Figure 5.

The experiments were conducted for the following temperature-dependent variables:

Thermal diffusivity [22]:α=0.0288×T+51.0646

α=0.0288·T+51.0646, $\left(\frac{m^{2}}{s}\right)$

where: T—material temperature at a mesh node (°C).

The Biot number: $B_{i}=\frac{\alpha \Delta x}{\lambda }$,

where: Δx—distance between two mesh nodes (m),

Fourier number: $F_{o}=\frac{\lambda \Delta t}{\Delta x}$,

where: Δt—time elapsed between the subsequent steps (s).

Thermal conductivity: $\lambda =\alpha \rho c_{p}$, $\left(\frac{W}{m{}^{\circ}C}\right)$

The numerical analysis of the distribution of temperature fields in the workpiece was performed along the axis of symmetry, where the thermocouples were mounted. The arrangement of the thermocouples is illustrated in Figure 6.

## 4. Results and Discussion

The quantities measured were the changes in the workpiece temperature below the surface being milled. The temperature was monitored by six thermocouples placed in the workpiece cross-section, as shown in Figure 6.

Table 2 shows the maximum temperature registered for each milling strategy. The highest values of the workpiece temperature were reported for strategy 1, while the lowest for strategy 4. This suggests that the temperature is directly dependent on the process intensity. The smaller the depth of cut, the less heat is generated. This explains why in strategy 4, where the smallest allowance was left, the temperature was the lowest. Thermocouples 1 and 4 were mounted just below the place where the milling cutter entered the workpiece. Thermocouples 3 and 6 measured the workpiece temperature below the place of the tool exit. The highest maximum temperature was observed at the final stage of the cutting process, i.e., at the end where thermocouples 3 and 6 were fitted. This was due to the relatively high thermal conductivity and thermal capacity of the workpiece material allowing the heat to be stored inside.

Changes in the workpiece temperature were registered during the whole cutting process. Figure 7 shows the relationship between the workpiece temperature and the cutting time for the first milling strategy, which required removing a layer of 3 mm in one pass of the tool. As can be seen, the highest temperature of the workpiece (111.4 °C) was recorded by thermocouple 3 placed furthest away from the tool entrance into the workpiece material. The phenomenon of heat accumulation taking place inside the workpiece material is clearly visible in the diagram. Once the tool entered the material, the temperature of the workpiece rose gradually, which was registered by each subsequent pair of thermocouples fixed along the milling direction.

Figure 8 shows the changes in temperature of the workpiece milled in two passes of the cutting tool. Each group of peaks on the temperature−time curve corresponds to one pass. Different colours represent the results obtained from the different thermocouples. There is a step-like increase in temperature visible for each subsequent thermocouple, resulting from the cutting time and the thermophysical properties of the workpiece material. Since the milling cutter was in constant contact with the workpiece material, a certain amount of heat generated during the cutting process was continually transferred to the workpiece material, where it accumulated. The temperature indicated by thermocouples 2 and 3 was thus higher than that shown by thermocouple 1. The same observation was made for thermocouples 5 and 6 in relation to thermocouple 4 (Figure 6). The values of temperature registered during pass 2 were higher because the workpiece temperature before pass 2 was higher by a value of heat delivered to the workpiece during pass 1.

The workpiece temperature decreased between the two passes of the tool as heat was distributed in the whole volume of the material and the workpiece was cooled by the surrounding air. The temperature measured by thermocouple 3 dropped from 74.7 to 33.6 °C. After pass 2, the maximum temperature of the workpiece at the same measuring point was reported to rise to 86.4 °C.

The increase in the maximum temperature recorded by the subsequent thermocouples can be explained by analyzing the distribution of the temperature field in the longitudinal cross-section. An example distribution of temperature in the workpiece generated during strategy 1 at Δt = 5.2 s is shown in Figure 9. Special attention should be paid to the direction in which the source of heat moved and the distribution of temperature in the workpiece.

As can be seen, the area where the maximum temperature occurred was exactly above the second pair of thermocouples (thermocouples 2 and 5). Then, the source of heat moved left towards thermocouples 3 and 6. The region of higher temperature followed the heat source and soon thermocouples 3 and 5 would register, first, a sharp increase in temperature and then a significant decline. An example temperature distribution for Δt = 6.8 s is illustrated in Figure 10. Here, the heat source was above thermocouples 2 and 5, and before thermocouples 3 and 6.

A short rise in temperature was observed for each pass of the tool and its intensity was directly dependent on the intensity of the cutting process and the thermophysical properties of the material being milled. This indicates that when the thickness of the layer removed is smaller, the maximum temperatures registered by the thermocouples are lower.

Figure 11 and Figure 12 show the relationship between temperature and time for the next two milling strategies, i.e., strategies 3 and 4, respectively. Figure 11 illustrates the process of cutting performed in three passes, to a depth of 1 mm each, while Figure 12 depicts the process of material removal in six passes each to a depth of 0.5 mm.

For the strategy involving three passes of the cutting tool (Figure 11), the workpiece temperature measured by thermocouple 3 changed in the following way. During the first pass, it increased from an initial value of 24 °C to 60.9 °C. Then, it dropped to 31.7 °C. It increased again to 69 °C during the second pass. After that, it decreased to 36.5 °C and during the third pass, it reached a maximum value of 78.2 °C.

In the last milling strategy, there was a similar increase in the workpiece temperature caused by the subsequent passes of the tool (Figure 12). The maximum temperatures measured by thermocouple 3 in the six passes were as follows: 44.7, 49.8, 51.8, 55.8, 57.9 and 63.5 °C.

The maximum values of the workpiece temperature registered by thermocouple 3 during all the milling strategies are given in Table 3. The results suggest that the thinner the layer of the material removed (and accordingly, the greater the number of passes), the lower the maximum temperature of the workpiece. For instance, the difference in temperature measured by thermocouple 3 between strategy 1 and strategy 2 (one pass to a depth of 3 mm vs. two passes to a depth of 1.5 mm) is 22.4%, while the difference in temperature between strategy 3 and strategy 4 (three passes to a depth of 1 mm vs. six passes to a depth of 0.5 mm) is 18.8%. Although the difference was dependent on the cutting parameters and the thermophysical properties of the workpiece material, it fluctuated around 20%.

The 2D simulations of the heat flow were carried out in the workpiece cross-section along the axis of symmetry because the tips of the thermocouples were located in this plane (Figure 6). That made it possible to compare the simulation data with the experimental results. During the tests, the face milling tool travelled in the same direction in each pass, regardless of the number of passes. For this reason, the influence of the tool on the surface texture generated during climb milling or up milling was not considered.

Except for the analysis of the workpiece temperature, the tests involved assessing the surface flatness. The flatness was measured at selected points using a ZEISS Contura G2 RDS coordinate measuring machine (Carl Zeiss AG, Oberkochen, Germany).

Figure 13a shows the results obtained for the first milling strategy, when the required amount of material, i.e., a layer of 3 mm, was removed in one pass. In this case, the flatness deviation was 6.7 μm, and the value represented the largest distance between the reference planes. The maximum values of flatness deviation were reported at both ends of the workpiece, which corresponded to the beginning and end of the cutting process. Centrally, there was a clear depression. Figure 13b–d illustrates the workpiece surface flatness data for the other three strategies of cutting.

The flatness deviation was 10.9 μm after two passes, 11.1 μm after three passes, and 6 μm after six passes.

From the comparative analysis of the flatness data (Figure 13), it is apparent that the distribution of flatness deviation was similar for all the cutting strategies. In all the four cases, the deviations were much smaller in the central part of the workpiece top surface. This phenomenon can be explained by the fact that the conditions at one end of the workpiece differed from those at the other end or in the central area. Other important factors that might have affected the surface flatness included the thermophysical properties of the material and the workpiece geometry. It should be noted that the deviation (including the maximum deviation) decreased with the decreasing depth of cut. The flatness deviation is likely to be affected by the thermophysical properties of the workpiece material such as thermal conductivity and thermal capacity. Another vital property is definitely the thermal expansion of the workpiece material because its value rises proportionally to an increase in temperature. The coefficient of linear thermal expansion obtained for the aluminum alloy was 23.6 × 10^−6^ K^−1^ (Table 1).

The heat accumulation observed at the end of the workpiece and its effect on the distribution of flatness was described using the results of the numerical simulation. Changes in the shape of the temperature fields were analyzed for the final moment of the cutting process. Figure 13a–d depicts temperature fields generated for Δt = 7.8 s, 8.8 s, 9.8 s and 10.8 s, respectively. As can be seen from Figure 13a,b, the maximum temperature region follows the source of heat, when the process is stable, which means the region moves with the movement of the tool−workpiece contact zone. The red arrows in the figures indicate heat transfer direction. However, when the contact zone reaches the workpiece end, the direction in which the face of the high temperature region spreads changes (Figure 13c).

Not being able to propagate before the face of the high temperature field, the heat transfer is stopped at the end of the workpiece. The thermal energy builds up in the corner in contact with the tool and the heat flux changes its direction, bridging to the adjacent areas where the temperature is lower (Figure 14c). The inverse flow of temperature is perfectly illustrated for Δt = 10.8 s in Figure 14d. The tool is then no longer in contact with the workpiece. As can be seen, the accumulated heat flows from the cutting tool to the inside of the workpiece. When heat reaches the workpiece end, it is accumulated there and causes the material to expand.

High thermal expansion of the workpiece is assumed to be the main cause of the thermal deformations, which in turn, are directly responsible for the flatness deviation of the surface milled. The thermoelastic deformations of the workpiece determined by means of Ansys Workbench are depicted in Figure 15.

The analysis of the temperature fields shown in Figure 14 and the thermoelastic deformations illustrated in Figure 15 suggest that the characteristic nonlinear distribution of surface flatness is a consequence of spatial changes in the distribution of temperature. As can be seen, the value of flatness deviation seems to be directly dependent on the intensity of the cutting process, i.e., the process parameters, and the thermophysical properties of the workpiece material.

In this study, it was assumed that different flatness deviations would be reported for different machining strategies because of the differences in the value of heat flux. The numerical analysis (Figure 15) revealed that the central depression was due to changes in the temperature field. The measurement data also indicated that the highest flatness deviation was in the centre, which suggests good agreement of the measurement results with the simulation data.

Similar research, described in [18], was conducted for external longitudinal turning. The authors illustrated the evolution of the workpiece geometry and temperature field at three different moments. They used the transient temperature, i.e., the temperature the workpiece reached after roughing, to determine its thermoelastic deformations (geometry). They concluded that thermal displacements differ depending on the milling strategy, and therefore, the total cutting time and the rate of heat transfer. Another of their key findings was that the finishing allowance should be greater than the thermoelastic deformation expected to take place.

## 5. Results Validation

The measurement data from thermocouples 4, 5 and 6 were compared with the numerical results obtained for the same thermocouples using FEM-based Ansys Workbench.

The temperature values obtained with the mathematical model were registered at points corresponding to points at which the actual thermocouples were fitted. The points were determined using appropriate geometrical coordinates.

The plots for thermocouples 4 and 5 in Figure 16 show that the difference was approximately 10 °C. This was due to the dynamically moving heat source and the thermal inertness of the analyzed system. For thermocouple 6, the changes in temperature in the cross-section are represented more accurately because the device was located closer to the workpiece edge.

From the FEM simulation data in Figure 9, it was clear that in the places where thermocouples 4, 5 and 6 were installed (Figure 6), the numerically calculated temperature was similar to that obtained by measurement. For thermocouples 1, 2 and 3, however, the measured temperature differed from the simulated temperature by 20 °C because of the dynamic displacement of the heat source and the thermal inertness of the system tested.

## 6. Conclusions

The analysis of the experimental and numerical data reveals that the milling strategy has a considerable effect on the geometric and dimensional accuracy of the final product. Increasing the number of passes reduces the heat flux at the tool−workpiece interface, which causes a decrease in temperature in the entire thermodynamic system. This contributes to lower thermoelastic deformation and higher dimensional and geometrical accuracy of the workpiece. Knowledge of this relationship is particularly important when high-precision products are manufactured. Although the experiments were conducted for AlCu4MgSi (EN AW-2017) aluminum alloy, the findings can be used to predict the behavior of other aluminum-based materials because the physical phenomena occurring in the milling process are very similar in nature. The comparative study of the experimental and numerical results indicates that the area of the depression on the workpiece top surface coincides with the area of maximum thermoelastic deformations. This article confirms that the number of passes used to remove the required amount of material has a significant effect on the maximum temperature of the workpiece, and consequently, its surface properties. The source of heat in milling directly affects the distribution of temperature in the workpiece, causing its thermoelastic deformations; indirectly, it affects the surface texture, i.e., roughness, waviness and form. Our future studies will focus on the effect of the cutting temperature on the shape of the tool path where it enters and exits the workpiece.

## Figures and Tables

**Figure 1 materials-13-04542-f001:**
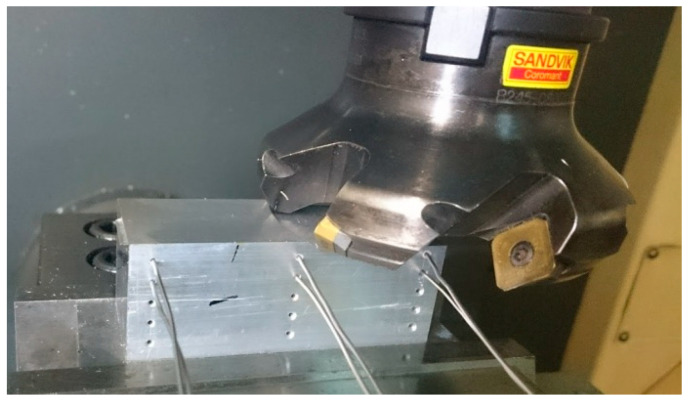
Experimental setup.

**Figure 2 materials-13-04542-f002:**
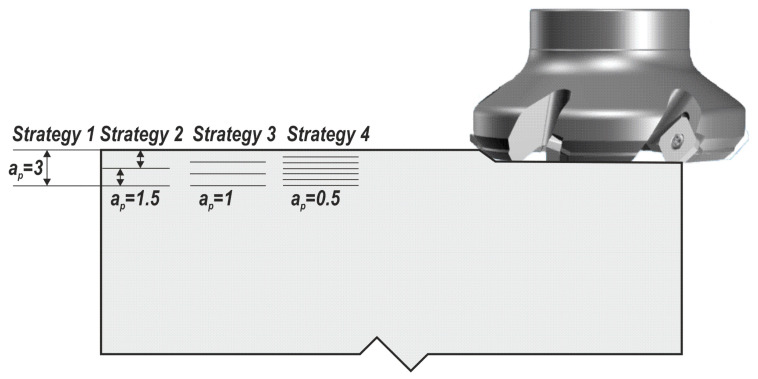
Cutting strategies.

**Figure 3 materials-13-04542-f003:**
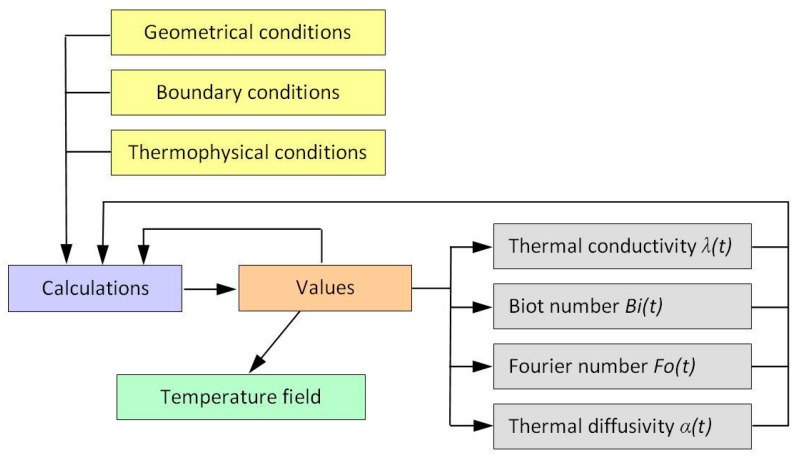
Calculation algorithm in the finite difference method (FDM).

**Figure 4 materials-13-04542-f004:**
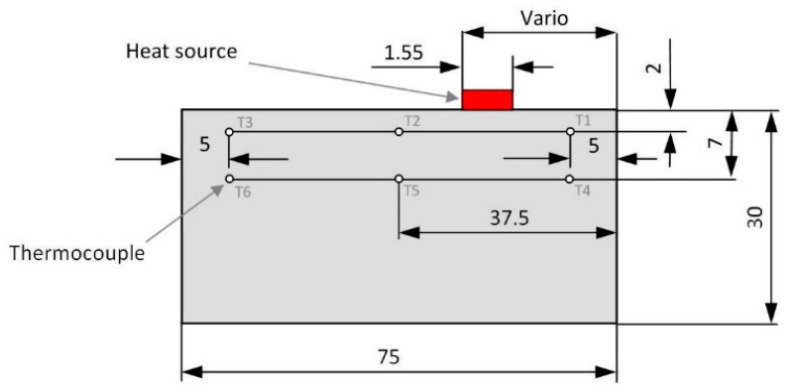
Calculation model generated using the FDM.

**Figure 5 materials-13-04542-f005:**
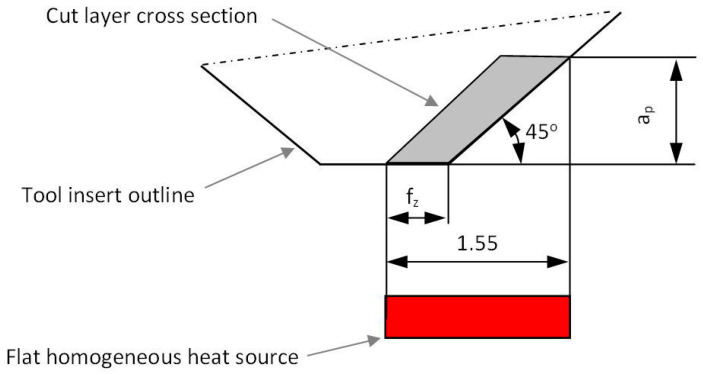
Determining the length of the flat source of heat for the first cutting strategy.

**Figure 6 materials-13-04542-f006:**
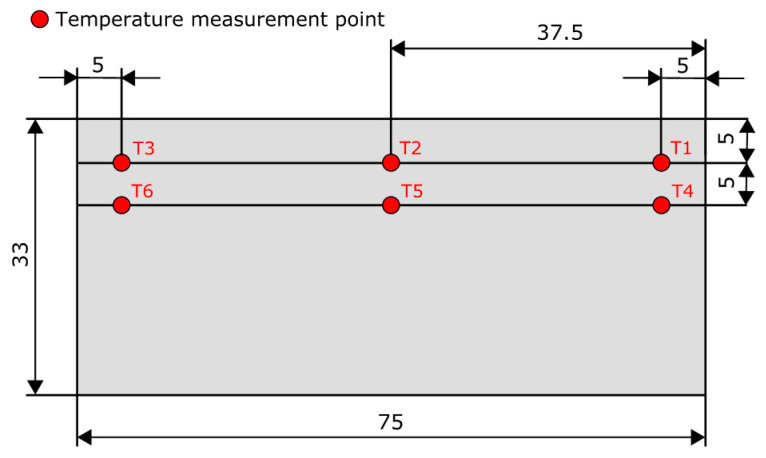
Arrangement of thermocouples in the workpiece cross-section.

**Figure 7 materials-13-04542-f007:**
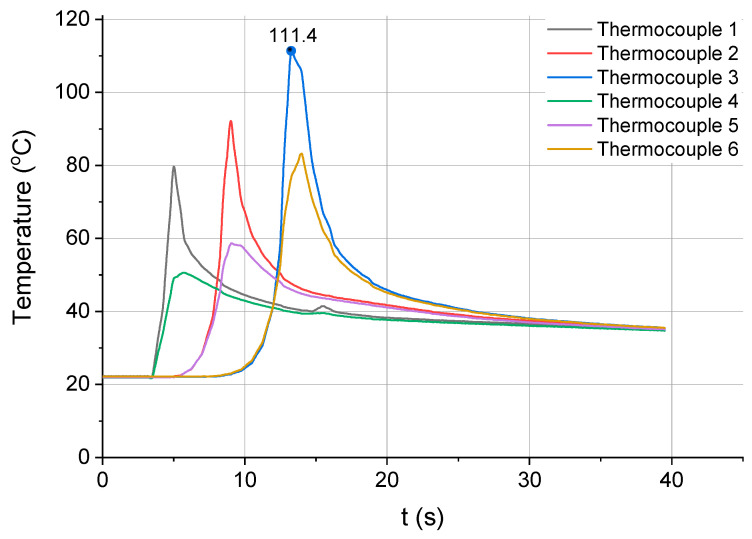
Workpiece temperature measured for strategy 1 (one pass with a_p_ = 3mm).

**Figure 8 materials-13-04542-f008:**
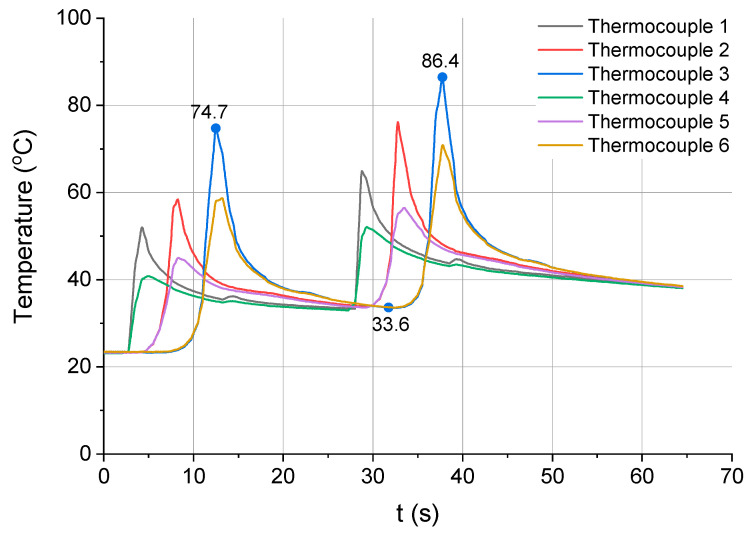
Temperature vs. time for strategy 2 (two passes, a_p_ = 1.5 mm).

**Figure 9 materials-13-04542-f009:**
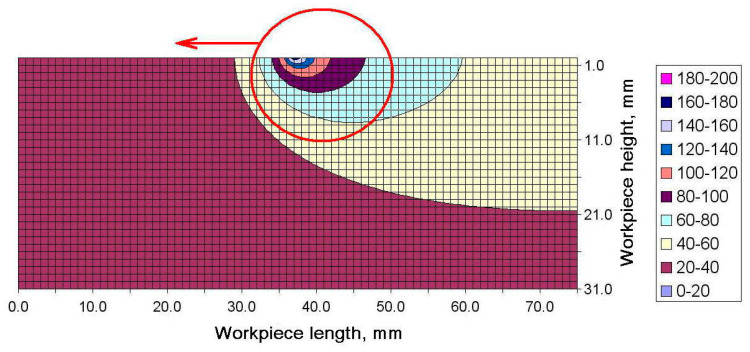
Temperature field registered for milling strategy 1 after Δt = 5.2 s.

**Figure 10 materials-13-04542-f010:**
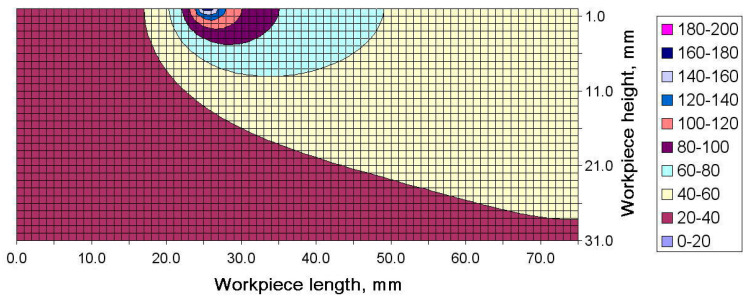
Temperature field for milling strategy 1 after Δt = 6.8 s.

**Figure 11 materials-13-04542-f011:**
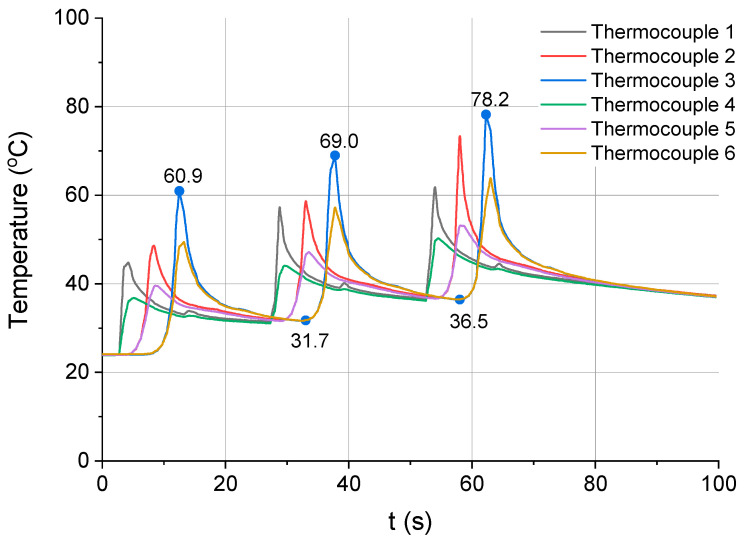
Temperature vs. time for the third milling strategy (three passes of the tool with ap = 1 mm).

**Figure 12 materials-13-04542-f012:**
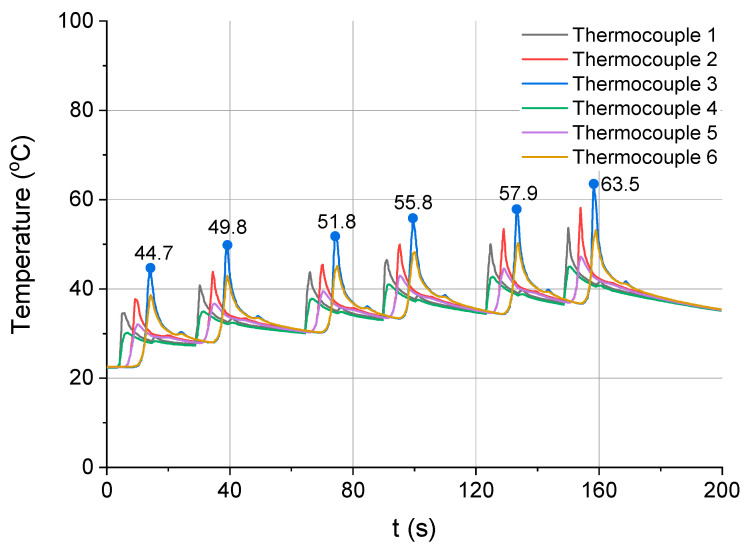
Temperature vs. time for strategy 4 (six passes of the tool with a_p_ = 0.5 mm).

**Figure 13 materials-13-04542-f013:**
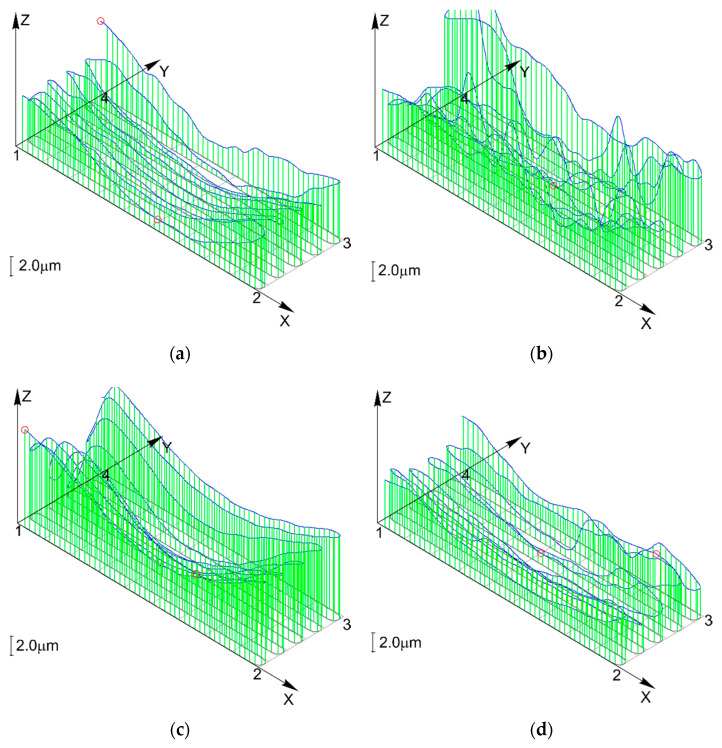
Surface flatness for (**a**) strategy 1, (**b**) strategy 2, (**c**) strategy 3 and (**d**) strategy 4.

**Figure 14 materials-13-04542-f014:**
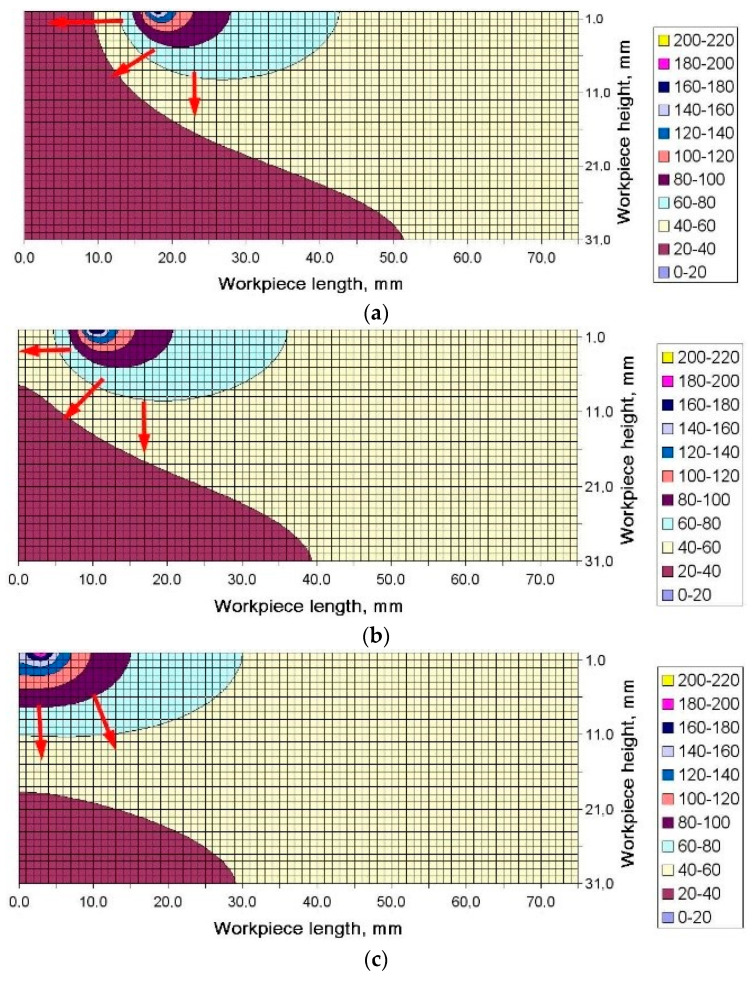
Temperature distribution for the first milling strategy in the cross-section along the workpiece length at (**a**) Δt = 7.8 s, (**b**) Δt = 8.8 s, (**c**) Δt = 9.8 s, and (**d**) Δt = 10.8 s.

**Figure 15 materials-13-04542-f015:**
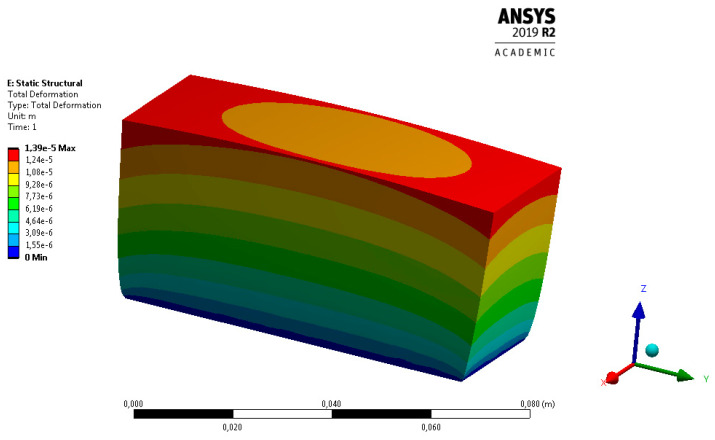
Total thermal deformations of the workpiece.

**Figure 16 materials-13-04542-f016:**
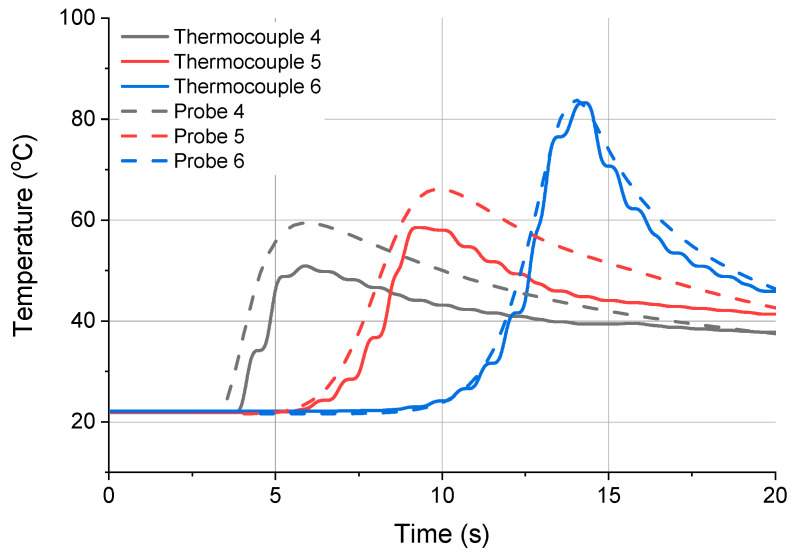
Measurement results compared with the numerical simulation data obtained using Ansys Workbench.

**Table 1 materials-13-04542-t001:** Properties of the aluminum alloy (EN AW-2017) solution treated and normalized [21].

Property	Value
Offset yield strength, R_p0.2_	275 MPa
Ultimate tensile strength, R_m_	427 MPa
Hardness, HB	105
Density, ρ	2790 kg·m^−3^
Coefficient of linear thermal expansion, τ	23.6 × 10^−6^ K^−1^
Specific heat, c_p_	873 J kg^−1^ K^−1^

**Table 2 materials-13-04542-t002:** Maximum temperatures during four different cutting strategies at six different measuring points.

Thermocouples	Temperature °C in Different Strategies			
	1	2	3	4
Thermocouple 1	79.5	64.8	61.8	53.7
Thermocouple 2	92.2	75.9	73.4	58.2
Thermocouple 3	111.4	86.4	78.2	63.5
Thermocouple 4	51.0	52.0	50.0	45.0
Thermocouple 5	58.5	56.5	53.1	47.1
Thermocouple 6	83.3	70.9	63.9	53.2

**Table 3 materials-13-04542-t003:** Maximum temperature registered by thermocouple 3.

Pass Number	Strategy 1	Strategy 2	Strategy 3	Strategy 4
Pass 1	111.4 °C	74.7 °C	60.9 °C	44.7 °C
Pass 2		86.4 °C	69.0 °C	49.8 °C
Pass 3			78.2 °C	51.8 °C
Pass 4				55.8 °C
Pass 5				57.9 °C
Pass 6				63.5 °C

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
