# Peer review of "Influence of the Cutting Strategy on the Temperature and Surface Flatness of the Workpiece in Face Milling"

_materials, 2020, doi:10.3390/ma13204542_

Round 1

Reviewer 1 Report

  • The heat generated by multiple milling passes of smaller thickness vs the heat generated by single or double passes is known, what does this article bring to the field that is not already common sense?
  • The abstract mentions an inappropriate set of cutting parameters, yet according to lines 82-83 only one value for feed and rotational speed have been used, how and why did the authors fix these parameters at that stage?
  • Line 129 the spacing in the size is unclear (did they authors mean 25 10-4 mm² of area? like 5 micronsx 5 microns).
  • Line 304 the authors mention the thermocouples 1, 3 and 6 but the figure below (Figure 15) shows comparison between thermocouples 4, 5 and 6 instead. And there is no discussion about the comparison of the FE simulation and experimental data nor any comment about the figure. Can the authors comment on the figure for section 6?
  • The increase in the number if passes does reduce the temperature for each pass but does increase the time for operation and thus puts a long steady wear on the toolpiece which can decrease the machining quality by wear so it is not a straight forward conclusion. The authors should at least attempt to make a comment about this added phenomenon in the conclusions/discussions and not just the introduction.
  • What do the author think about using air cooling during machining?
  • In the conclusions, the authors state that the findings can be applied to other materials, but contrarily to Steel and stainless steel, aluminum is relatively easy to machine and the heat generation during milling is relatively low. I would be more cautious as to make such an affirmation.
  • It seems that the highest temperature reached by the Al piece was 111.4°c which is still below Debye’s temperature for Al and far from the melting temperature, as a consequence what surface properties is expected to be affected?

Author Response

Response to the reviewers’ comments

Reviewer #1: Review on '' Influence of the cutting strategy on the temperature and surface flatness of the workpiece in face milling ''.

Rev.: The heat generated by multiple milling passes of smaller thickness vs the heat generated by single or double passes is known, what does this article bring to the field that is not already common sense?

Temperature measurement of the workpiece is extremely important due to the phenomenon of thermoelasticity and its thermal expansion. The material has isotropic properties which make it expand freely in all directions during machining. This affects on the dimensional and geometrical accuracy of the products. This phenomenon is particularly important during e.g. roughing machining (with increased cutting parameters) without adding cooling and lubricating liquids. Due to the factors, the experiment was carried out for different depths of cut with constant feed per tooth and spindle speed.

Rev.: The abstract mentions an inappropriate set of cutting parameters, yet according to lines 82-83 only one value for feed and rotational speed have been used, how and why did the authors fix these parameters at that stage?

The sentence in the abstract has been rebuilt. The technological parameters such as cutting speed and feed rate per tooth were selected based on the recommendations of the manufacturer of the cutting inserts. Cutting tool manufacturers generally do not provide recommended depths of cut. The experiment was carried out for four different values of the depth of cut.

Rev.: Line 129 the spacing in the size is unclear (did they authors mean 25 10-4 mm² of area? like 5 micronsx 5 microns).

Indeed, the notation used in the article may be fallible. Square elements with a side length of 2.5 micrometers were used in the calculations. The value of 0.0025 mm given in the article means the length of the side of a square mesh in the calculation grid.

Rev.: Line 304 the authors mention the thermocouples 1, 3 and 6 but the figure below (Figure 15) shows comparison between thermocouples 4, 5 and 6 instead. And there is no discussion about the comparison of the FE simulation and experimental data nor any comment about the figure. Can the authors comment on the figure for section 6.

The assignment of thermocouple numbers has been corrected. In section 6 text relating to the validation of the results of the FE simulation in relation to temperature measurement with thermocouples has been added. Analyzing the images (Fig. 8) obtained from the FE simulation, it was noticed that in the places where the thermocouples 4, 5 and 6 are located (Fig. 5), the temperature from the numerical calculations is consistent with the measurement results. For thermocouples 1, 2 and 3, the registered  temperature differs from the temperature obtained on the basis of the simulation by 20 ° C due of the dynamic displacement of the heat source and the thermal inertness of the tested system.

Rev.: The increase in the number if passes does reduce the temperature for each pass but does increase the time for operation and thus puts a long steady wear on the toolpiece which can decrease the machining quality by wear so it is not a straight forward conclusion. The authors should at least attempt to make a comment about this added phenomenon in the conclusions/discussions and not just the introduction.

The comment was added to the conclusions.

Increasing the number of passes reduces the heat flux generated between the tool and the workpiece, which cause a reduction of the temperature in the entire thermodynamic system. The effect of this is the reduction of thermoelastic deformation of the workpiece, thanks to a higher the dimensional and geometrical accuracy. of the workpiece can be obtained. With this cutter head diameter and the type of cutting inserts and material properties to be machined, the wear of the cutting edge is of secondary importance.

Rev.: What do the author think about using air cooling during machining?

 Air cooling is indicated during milling, in particular of aluminum alloys, however, in order to recreate of the experiment with simulation conditions in cutting tests, cooling was abandoned.

Rev.: In the conclusions, the authors state that the findings can be applied to other materials, but contrarily to Steel and stainless steel, aluminum is relatively easy to machine and the heat generation during milling is relatively low. I would be more cautious as to make such an affirmation.

We agree. The sentence has been rebuilt.

Rev.: It seems that the highest temperature reached by the Al piece was 111.4°c which is still below Debye’s temperature for Al and far from the melting temperature, as a consequence what surface properties is expected to be affected?

The source of heat acting during the milling process affects the temperature distribution, which causes thermoelastic deformation of the machined object, which indirectly affects the flatness of the formed surface, its waviness and shape errors.

Reviewer 2 Report

Review: Influence of the cutting strategy on the temperature and surface flatness of the workpiece in face milling

The authors proposed an interesting and difficult topic as measuring temperature in milling is. The conducted experimental and numerical results but the paper needs further improvement in order to be more solid.

  • Abstract: the authors did not address why it is important to measure temperature on the workpiece. The scope of the paper needs to be revised.
  • The state of art is weak with a poor number of references and the authors failed at explaining the lack of knowledge in the literature which is the reason for the paper.
  • Please add a figure or a table to explain the different strategies and what was the specific purpose of them…
  • Section 3 is weak. Those are known heat transfer equations. If the authors do not improve it and introduce further concepts, it could be removed.
  • What is the purpose of adding Figure 14. It seems more a qualitative figure with low connection with the previous information and vaguely detailed.
  • It is difficult to understand Figure 12 as presented. It lacks of quality (typo too small) and there are not any variables on the different axes….
  • Figure 15 is related to thermocouples 1, 3 and 6 (as the text says) or with 4 , 5 and 6 (as the legend in the figure).
  • There is not a true validation of the numerical results with the experimentally obtained data from the thermocouples. The reader expects a correspondence between both (maybe in Section 6?). If not, how can the authors ensure a useful method?

There are a number of missed/neglected things, which are valuable for a paper that sounds.

  • The numerical model does not consider that the tool is a rotating heat source? We have a 2D model and results will be the same for down- or up-milling conditions.
  • Other important cutting data: number of flutes of the tool (which was not considered in the simulation), ae (which it is supposed to be 33mm?), Vc (that can be derived but is essential to mechanisms related with temperature)

Author Response

Response to the reviewers’ comments

Reviewer #2: Review on '' Influence of the cutting strategy on the temperature and surface flatness of the workpiece in face milling ''.

The authors proposed an interesting and difficult topic as measuring temperature in milling is. The conducted experimental and numerical results but the paper needs further improvement in order to be more solid.

Rev.:  Abstract: the authors did not address why it is important to measure temperature on the workpiece. The scope of the paper needs to be revised

The abstract has been supplemented with the required information. Temperature measurement of the workpiece is extremely important due to the phenomenon of thermoelasticity and its thermal expansion. The material has isotropic properties which make it expand freely in all directions during machining. This affects on the dimensional and geometrical accuracy of the products (the machining was carried out without the cooling and lubricating liquid).

Rev.: The state of art is weak with a poor number of references and the authors failed at explaining the lack of knowledge in the literature which is the reason for the paper?

On the basis of the literature review, it was not possible to find  in scientific studies enough information about the impact of thermal expansion on the dimensional and geometrical accuracy of objects made by machining. There is insufficient information on the influence of thermoelasticity on the dimensional and geometrical errors milling objects, which is the reason for the article.

Rev.:  Please add a figure or a table to explain the different strategies and what was the specific purpose of them…

The figure was added to the article.

Rev.:  Section 3 is weak. Those are known heat transfer equations. If the authors do not improve it and introduce further concepts, it could be removed.

The equations have been removed from the article and replaced by a comment: " In heat calculation programs using numerical simulation methods known equations governing the phenomena of heat transfer have been implemented".

Rev.: What is the purpose of adding Figure 14. It seems more a qualitative figure with low connection with the previous information and vaguely detailed.

Figure 14 is the result of a simulation carried out in ANSYS software and shows the thermal deformation of the tested element under the influence of spatial temperature distribution. Figure 14 is related to Figure 12 because it shows the characteristics of thermoelastic deformation in relation to the measured flatness deviation of the face of the milled element.

Rev.: It is difficult to understand Figure 12 as presented. It lacks of quality (typo too small) and there are not any variables on the different axes….

The figure has been corrected.

Rev.:  Figure 15 is related to thermocouples 1, 3 and 6 (as the text says) or with 4 , 5 and 6 (as the legend in the figure).

Corrected.

Rev.: There is not a true validation of the numerical results with the experimentally obtained data from the thermocouples. The reader expects a correspondence between both (maybe in Section 6?). If not, how can the authors ensure a useful method?

The assignment of thermocouple numbers has been corrected. In section 6 text relating to the validation of the results of the FE simulation in relation to temperature measurement with thermocouples has been added. Analyzing the images (Fig. 8) obtained from the FE simulation, it was noticed that in the places where the thermocouples 4, 5 and 6 are located (Fig. 5), the temperature from the numerical calculations is consistent with the measurement results. For thermocouples 1, 2 and 3, the registered  temperature differs from the temperature obtained on the basis of the simulation by 20 ° C due of the dynamic displacement of the heat source and the thermal inertness of the tested system.

There are a number of missed/neglected things, which are valuable for a paper that sounds.

Rev.:  The numerical model does not consider that the tool is a rotating heat source? We have a 2D model and results will be the same for down- or up-milling conditions.

2D simulations of heat flow in the workpiece were made for the sample cross-section located exactly along the sample axis. This was done because the tips of the measuring thermocouples were located in this plane (Fig. 5). In this way, it is able to compare experimental and simulation results. During the experiment, the face milling tool was guided in the same direction in each pass, regardless of the number of passes. For this reason, the influence on the final result of climb milling or up milling has not been considered.

Rev.:  Other important cutting data: number of flutes of the tool (which was not considered in the simulation), ae (which it is supposed to be 33 mm?), Vc (that can be derived but is essential to mechanisms related with temperature)

The heat flux value was calculated based on the formula

where: Fc – cutting force N, vc – cutting speed m/s.

The value of cutting force Fc can be found using an well known equation.

The simplifications used in the calculations do not result from the lack of knowledge and experience, but from the fact that they were 2D calculations for the cross-section located exactly along the axis of the sample. The face milling tool was also moving along the same axis. Such an arrangement in one plane significantly simplifies the calculation procedure

Reviewer 3 Report

The paper presents a case study on face milling of aluminium and the effect of milling strategy on temperature and resulting workpiece geometry. The paper is quite interesting and well presented.

However, there are some issues that should be dealt with before the paper is ready for publication.

Section 1:

  1. The introduction is very general and treats machining and milling too imprecisely. The paper would benefit from focusing on a specific application and present the relevant background literature for that.
  2. The objective of the study is not clearly stated. I expect to find it at the end of the introduction section.
  3. How other literature is referred to in the introduction needs to be more consistent. For example, see lines 45-49. References 6-8 are cited in a sweeping manner and then the content of one of the references (7) is explained but not the other (6 and 8).

Section 2:

  1. It is not explained why this particular experimental setup was selected. Why aluminium? Why face milling? See my point 1 above. The application should be introduced and motivated in the introduction section.
  2. The selection of the 4 different milling strategies is not explained. Are they representative of different real world cases? How was the selection of those strategies made?

Section 5:

  1. The results are not related to the literature and the open research questions that should be found in section 1 (see my points 1-3 above). How do the results compare to what was expected and to what others have found?

Section 6:

  1. In figure 15 there is a difference in temperature between thermocouples and probes 4 and 5 of about 10 degrees respectively, but not for number 6. This difference is not discussed. Why is the model giving the correct temperature for one position but not two other positions? Also, is the numbering correct? It does not match the text above the figure (lines 304-305).
  2. Figure 15 is not referred to in the text.

Author Response

Response to the reviewers’ comments

Reviewer #3: Review on '' Influence of the cutting strategy on the temperature and surface flatness of the workpiece in face milling ''.

The paper presents a case study on face milling of aluminium and the effect of milling strategy on temperature and resulting workpiece geometry. The paper is quite interesting and well presented.

However, there are some issues that should be dealt with before the paper is ready for publication.

Section 1

Rev.:  The introduction is very general and treats machining and milling too imprecisely. The paper would benefit from focusing on a specific application and present the relevant background literature for that

The introduction has been supplemented with additional references. On the basis of the literature review, it was not possible to find  in scientific studies enough information about the impact of thermal expansion on the dimensional and geometrical accuracy of objects made by machining. There is insufficient information on the influence of thermoelasticity on on the dimensional and geometrical errors milling objects, which is the reason for the article.

Rev.: The objective of the study is not clearly stated. I expect to find it at the end of the introduction section.

The objective of the study was added at the end of the introduction session. The objective of the study was to determine how the heat source during the milling process affects on the temperature distribution and thermoelastic deformation of the machined material. An additional goal was to determine the effect of the machining  strategy on the flatness of the surface.

Rev.:  How other literature is referred to in the introduction needs to be more consistent. For example, see lines 45-49. References 6-8 are cited in a sweeping manner and then the content of one of the references (7) is explained but not the other (6 and 8).

References to items 6 and 8 from the literature have been expanded.

Section 2:

Rev.:  It is not explained why this particular experimental setup was selected. Why aluminium? Why face milling? See my point 1 above. The application should be introduced and motivated in the introduction section.

The note was added to the introduction. The face milling process with tools equipped in inserts generates a lot of heat during the process. Face milling is often the final machining that shapes the surface and its usable properties. In the article, it was decided to check how the temperature for various strategies affects on dimensional and geometrical errors of objects in the milling process. The processed material is aluminum because it is one of the most popular alloys used in the industry, e.g. automotive or aviation.

Rev.: The selection of the 4 different milling strategies is not explained. Are they representative of different real world cases? How was the selection of those strategies made?

Cutting tool manufacturers generally do not provide recommended depths of cut. The technologist's task is to select the appropriate depth of cut and other parameters to meet the dimensional and geometry requirements for the workpiece. The selection of the cutting depth is particularly important in e.g. roughing and finishing operations (with increased cutting parameters) without supplying cooling and lubricating liquids as it affects on  the accuracy of the manufactured items.

Section 5:

Rev.: The results are not related to the literature and the open research questions that should be found in section 1 (see my points 1-3 above). How do the results compare to what was expected and to what others have found?

The expected result of the research was to obtain different flatness deviations as a result of using different machining strategies due to different heat flux values. After conducting a numerical simulation showing the deformation of the tested element under the influence of temperature (Fig. 14), the concave area in the central area of the sample was expected. Based on the measurements of the surface flatness, it was noticed that the minimum flatness deviation is in the middle of the plane, which is consistent with the simulation.

Section 6:

Rev.:  In figure 15 there is a difference in temperature between thermocouples and probes 4 and 5 of about 10 degrees respectively, but not for number 6. This difference is not discussed. Why is the model giving the correct temperature for one position but not two other positions? Also, is the numbering correct? It does not match the text above the figure (lines 304-305).

The assignment of thermocouple numbers in lines has been corrected. In Figure 15, for thermocouples 4 and 5, the difference is approximately 10 ° C. This is due to the dynamically moving heat source and the thermal inertness of the measuring system. For the thermocouple 6 there is a more accurate representation of the temperature course due to its location at the edge of the sample, which is associated with the sample heating up to the same temperature throughout the entire section.

Rev.: Figure 15 is not referred to in the text.

Figure 15 has been commented on the text and assigned a reference.

Round 2

Reviewer 2 Report

Minor actions are required. Nomenclature should be extended or removed 

Author Response

Thank you for your suggestions, the nomenclature has been removed.

Reviewer 3 Report

The quality of the paper has been much improved in terms of being a research paper. E.g. the objective is clearly defined and motivations for choice of application to study and experimental setup is explained better.

However, minor improvements should still be made.

In the previous review report I wrote:

The results are not related to the literature and the open research questions that should be found in section 1 (see my points 1-3 above). How do the results compare to what was expected and to what others have found?

And the response was:

The expected result of the research was to obtain different flatness deviations as a result of using different machining strategies due to different heat flux values. After conducting a numerical simulation showing the deformation of the tested element under the influence of temperature (Fig. 14), the concave area in the central area of the sample was expected. Based on the measurements of the surface flatness, it was noticed that the minimum flatness deviation is in the middle of the plane, which is consistent with the simulation.

I do not see this explained as clearly and concisely as this in the paper. It could be stated more or less as above, in the response, in the end of section 4 as a sort of summary. It would also be nice to include something about how this is in line with, or not, with what can be expected after reading the literature cited in the introduction. Give one or a few examples of how this relates to the previous research.

One additional comment. While the use of English is ok it can be improved for the sake of readability. The structure of some sentences makes the text difficult to read.

Author Response

Dear reviwer, I hope I have been able to improve everything according to your suggestions.